# Exploring the Relationship between Biosynthetic Gene Clusters and Constitutive Production of Mycosporine-like Amino Acids in Brazilian Cyanobacteria

**DOI:** 10.3390/molecules28031420

**Published:** 2023-02-02

**Authors:** Rafael B. Dextro, Endrews Delbaje, Vanessa Geraldes, Ernani Pinto, Paul F. Long, Marli F. Fiore

**Affiliations:** 1Center for Nuclear Energy in Agriculture (CENA), University of São Paulo (USP), Avenida Centenário 303, Piracicaba 13416-000, Brazil; 2School of Pharmaceutical Sciences, University of São Paulo (USP), Avenida Prof. Lineu Prestes 580, São Paulo 05508-000, Brazil; 3Faculty of Life Sciences & Medicine, King’s College London, London SE1 9NH, UK

**Keywords:** mycosporine-like amino acids, cyanobacteria, biosynthesis, gene cluster evolution

## Abstract

Cyanobacteria are oxygenic phototrophic prokaryotes that have evolved to produce ultraviolet-screening mycosporine-like amino acids (MAAs) to lessen harmful effects from obligatory exposure to solar UV radiation. The cyanobacterial MAA biosynthetic cluster is formed by a gene encoding 2-*epi*-5-*epi*-valiolone synthase (EVS) located immediately upstream from an *O-*methyltransferase (OMT) encoding gene, which together biosynthesize the expected MAA precursor 4-deoxygadusol. Accordingly, these genes are typically absent in non-producers. In this study, the relationship between gene cluster architecture and constitutive production of MAAs was evaluated in cyanobacteria isolated from various Brazilian biomes. Constitutive production of MAAs was only detected in strains where genes formed a co-linear cluster. Expectedly, this production was enhanced upon exposure of the strains to UV irradiance and by using distinct culture media. Constitutive production of MAAs was not detected in all other strains and, unexpectedly, production could not be induced by exposure to UV irradiation or changing growth media. Other photoprotection strategies which might be employed by these MAA non-producing strains are discussed. The evolutionary and ecological significance of gene order conservation warrants closer experimentation, which may provide a first insight into regulatory interactions of genes encoding enzymes for MAA biosynthesis.

## 1. Introduction

Cyanobacteria are the only known prokaryotes to carry out oxygenic photosynthesis and play a crucial role as primary producers and nitrogen fixers in both aquatic and terrestrial habitats [1,2]. Obligatory requirements of phototrophy necessitate exposure to potentially damaging solar UV radiation [3]. To palliate harmful effects associated with solar UV light exposure, many cyanobacterial strains biosynthesize photoprotective and antioxidant substances such as mycosporine-like amino acids (MAAs) [4]. This family of over 30 colourless and water-soluble metabolites comprises either a cyclohexenimine or cyclohexenone UV-absorbing chromophore, conjugated to the nitrogen or imino-alcohol component of amino acids. MAAs have absorption maxima (λmax) in the UVB (280–315 nm) and UVA (315–400 nm) regions of solar radiation [5] and are surmised to afford photoprotection to producing organisms by dissipating absorbed solar radiation as heat energy [6].

The MAA biosynthetic pathway in some cyanobacteria is derived from the intermediate sedoheptulose 7-phosphate (SH 7-P) of the pentose phosphate pathway. The cyanobacterial SH 7-P cyclase (2-*epi*-5-*epi*-valiolone synthase, EVS), together with an *O-*methyltransferase (OMT), are the first and second enzymes in the pathway leading to the MAA precursor, 4-deoxygadusol [7]. Genome mining has revealed that the genes encoding EVS and OMT are prevalent in many cyanobacteria known to produce MAAs and, accordingly, these genes are typically absent in non-producers [8]. The gene arrangement found in the strain *Synechocystis* sp. PCC6803 is an exception. Even without EVS (*mys*A), this strain produced three novel MAAs following UV irradiation [9]. Additionally, when the EVS encoding gene was deleted from *Anabaena variabilis* ATCC 29413, the resulting mutant still produced the MAA shinorine at levels nearly equivalent to that of the wild genotype following UV irradiance [10]. Hence, the presence of an EVS encoding gene for MAA biosynthesis may not be essential in cyanobacteria, and it has been proposed that functionally duplicate yet distinct convergent pathways for the biosynthesis of MAAs might co-exist [11]. Nevertheless, genome mining and heterologous gene expression data provide compelling evidence for MAA biosynthesis originating from the pentose phosphate pathway to be widespread amongst cyanobacteria, and the expected genetic arrangement to produce MAAs has been described, which is linked to four genes arranged in a co-linear cluster [12,13]. The biosynthetic enzymes encoded by these genes are EVS (*mys*A), OMT (*mys*B), and an ATP-grasp ligase (*mys*C), followed immediately downstream by either a d-Ala-d-Ala ligase (*mys*D) or a non-ribosomal peptide synthetase (NRPS-like) gene (*mys*E) to complete the four genes cluster. These genes encode proteins to finish imino-MAA biosynthesis via separate ligase reactions [14,15,16].

From the foregoing evidence, it remained to be determined if there were further divides in the co-linear gene cluster arrangement for MAA biosynthesis relevant to photoprotective or other cellular functions of MAAs in cyanobacteria. Ten cyanobacteria were used for this study, all had previously been isolated from diverse tropical habitats characterized by year-round exposure to intense solar UV irradiation. Genome mining and phylogenetic analysis were used to investigate variations in co-linear gene cluster architecture in these cyanobacteria and to establish a link between the genetic context of MAA biosynthetic genes and measured constitutive and irradiated UV-induced MAA productions.

## 2. Results

### 2.1. Gene Cluster Manual Annotation

Manual annotation of genome sequences retrieved from the NCBI database revealed MAA biosynthetic genes either had a co-linear arrangement or the genes were separated with intragenomic spaces of >15 Kb (Figure 1). Only in *Nodularia spumigena* CENA596 and *Sphaerospermopsis torques-reginae* ITEP-024 the canonical co-linear *mys*ABCD cluster was found. In both *Brasilonema* strains, the cluster was found with a different gene configuration, formed by *mys*ABC followed by an additional methyltransferase, a transporter, a dioxygenase (*mys*H), and an NRPS-like (*mys*E). The remaining strains, all originally from the Pantanal biome (*Arthrospira platensis* CENA597, *Arthrospira platensis* CENA650, *Alkalinema pantanalense* CENA528, *Geminocystis* sp. CENA526, *Pantanalinema rosanae* CENA516, and *Anabaenopsis elenkinii* CCIBt3563), seem to possess non-clustered low identity genes related to MAA biosynthesis scattered throughout their genomes.

Each annotated gene had the expected functional motif of the corresponding enzyme related with the biosynthesis of MAAs (Appendix A). Identification was based on the similarity of the sequences with known genes deposited in NCBI (Appendix A).

### 2.2. Analysis of MAA and Carotenoid Content

The quantitative LC/MS-MS analysis of the 10 cyanobacterial strains is shown in Table 1 (detailed spectroscopic data of all biological replicates is shown in the Appendix A to highlight natural biological variation in titres). Concentrations of shinorine and porphyra-334 were detected in the extract of *Nodularia spumigena* CENA596 and *Sphaerospermopsis torques-reginae* ITEP-24. The extract of *Sphaerospermopsis torques-reginae* ITEP-024 contained a much higher total concentration of shinorine than *Nodularia spumigena* CENA596 strain, with greater absolute value detected in BG-11 (although without significant statistical difference from the other media tested). Conversely, the extract of CENA596 contained higher porphyra-334 content. Again, the absolute value measured in BG-11 was the highest (BG-11 > Z8 > ASM-1) but without significant statistical difference from Z8 and ASM-1. The extract of ITEP-024 also presented LC peak characteristics for mycosporine-glycine-alanine (MGA) when grown in BG-11 medium, and for palythine when grown in BG-11 and Z8 media. The palythine peak profile was also observed in both *Brasilonema* strains. The extracts of the remaining strains yielded no measurable production of the four MAAs for which standards were available and no peaks characteristic for MAAs could be seen following UV-visible spectroscopy (Appendix A).

As expected, the MAA production profiles of *Nodularia spumigena* CENA596, *Sphaerospermopsis torques-reginae* ITEP-24, and the two *Brasilonema* strains showed an increase in total MAA content after 72 h exposure to UVA+UVB lights (Table 2 and Appendix A). The concentration of porphyra-334 was 3.2 times higher in the extract of *Nodularia spumigena* CENA596 and 7.4 times higher in the extract of *Sphaerospermopsis torques-reginae* ITEP-024 following UV irradiance. The concentration of shinorine was 2.3 times higher in extracts of UV-irradiated cultures of ITEP-024. This strain also contained detectable levels of MGA whereas *Nodularia spumigena* CENA596 did not (Table 2). Extracts from both *Brasilonema* strains presented LC/MS-MS profiles consistent with palythine when compared with the standard compound, with an increase in the concentration of palythine in *Brasilonema octagenarum* UFV-OR1 (0.278 ± 0.039 μg/ mg) after UV light induction, being the highest concentration detected for this specific MAA in this study. Unexpectedly, no MAA production could be detected by LC/MS-MS or UV-visible spectroscopy in the extracts of the remaining cyanobacteria following UV irradiation (Table 2 and Appendix A).

The measured total carotenoid content varied from control to UV-exposed cultures (Appendix A). *Geminocystis* sp. CENA526 was the only strain with a significant increase after the 72 h irradiance period.

### 2.3. Phylogenetic Analyses

A maximum likelihood phylogenetic tree was constructed to illustrate evolutionary relationships between the cyanobacterial strains used in this study and data pertaining to MAA-related gene cluster architecture (Figure 2). From the strains used in the analysis, the presence of a co-linear MAA gene cluster was identified in the orders Nostocales, Chroococcales, and Pleurocapsales. Phylogenetic analysis of each gene related to MAA biosynthesis (Appendix A) showed two distinct groups. One clade contained strains with the biosynthetic genes organized into the expected co-linear cluster arrangement. The other group included strains where the MAA biosynthetic genes had a non-canonical arrangement and were scattered across these genomes. Individual gene order conservation was even more apparent when phylogenetic analysis of the entire canonical gene clusters was considered (Appendix A).

## 3. Materials and Methods

### 3.1. Cyanobacterial Strains

The ten strains used in this study are detailed in Table 3. The strains were grown in an appropriate Z8 liquid medium under fluorescent light (40–50 µmol photons·μm^−2^·s^−1^) operated in a 14:10 h light/dark cycle at 22 +/− 1 °C for 45 days before analysis of MAA content. Both *Brasilonema* strains, having slower growth rates than the other strains, were grown for an additional period of 45 days to achieve sufficient biomass for MAA extraction. The influence of culture media on titres of constitutive MAA production was also evaluated using BG-11 [17] and ASM-1 [18] for *Geminocystis* sp. CENA526, *Nodularia spumigena* CENA596, and *Sphaerospermopsis torques-reginae* ITEP-024. Besides Z8(0), *Brasilonema octagenarum* UFV-OR1 was grown only in BG-11 since no growth was observed in the ASM-1 medium. The compositions of these culture media are detailed in Appendix A. Biomass was concentrated by centrifugation (9000 rpm, 15 min, 5 °C) and lyophilized for MAA extraction.

### 3.2. MAA Induction via UVA+UVB

Cyanobacteria were grown for 42 days in the same culture media described in Table 3. The cultures were then exposed to UVA (using two Philips TL-K 40W/10R-UVA lamps providing 7.5 W/m^2^ of irradiance) and UVB lights (through two Philips TL 40W/12RS-UVB lamps providing 4.5 W/m^2^ of irradiance) for 72 h. To ensure efficient UV light exposure, the cultures were transferred to 250 mL clear optic polystyrene bottles with vented caps equipped with filters. MAA content was measured in irradiated cultures and compared with non-UV-exposed cultures that had been grown under cycles of fluorescent light for the same experimental period and in the same culture medium. Thus, the total experimental period for UV-exposed cultures was the same as those cultures which were not irradiated (both *Brasilonema* strains were grown for 87 days prior to the 72 h UV induction). The presence of MAAs was also inferred by absorbance of culture extracts between 310 to 360 nm.

### 3.3. Gene Cluster Manual Annotation

A gene reference library was created using MAA biosynthetic gene sequences deposited in the National Center for Biotechnology Information (NCBI) from *Anabaena variabilis* ATCC29413 (genes Ava_3855, 3856, 3857 and Ava_3858), *Nostoc punctiforme* ATCC29133/PCC 73102 (genes NpR5597, 5598, 5599 and NpR5600), and *Microcystis aeruginosa* PCC7806 (genes *mys*A, B, C, and *mys*D). These strains had previously been described as MAA producers in the literature [7,12,13,14]. The nucleotide sequences were translated and used as query sequences to search for additional MAA-related proteins in NCBI via online BLAST, adding new homologs with high sequence identity (>65%, e-value ≥ 1.10^−50^) to the reference library. When the sequence identity was lower, homology was confirmed using the Pfam database [25] and the Motif Search engine of the GenomeNet database in default settings [26,27]. The alignments for each gene were made with ClustalW. A final FASTA file that contained all the collated gene sequences was used to search for MAA-related genes in the experimental strains using BLAST + v2.2.29 [28,29], applying an identity cut-off of 60% and query coverage cut-off of 75%. For values of identity below the cut-off, motif analysis was again used to match similar protein domains. Additionally, manual annotation of the BLAST results was performed using the Artemis software [30] and a prediction of the translated protein sequence for each gene was made by searching the Pfam database. The manual annotation consisted of determining gene size and loci in relation to genes encoding proteins related to MAA biosynthesis. The sequences corresponding to the *mys* genes can be found in the GenBank database under the accession numbers shown in Appendix A.

### 3.4. MAA Extraction and Analysis

Approximately 5.0 mg lyophilized biomass of each sample, weighed in the precision analytical Ohaus AR3130 balance (Parsippany, NJ, USA), were extracted on ice in 2.0 mL 0.1% (*v*/*v*) formic acid and 0.2 mM ammonium formate (Buffer A), pH ~ 2.55 with disruption by one cycle of sonication (1 min per sample) and vortex mixing (5 s per sample). After 1 h at room temperature (~22 +/− 1 °C), the extracts were centrifuged (10,000 rpm, 10 min, 5 °C) and the supernatant was filtered to remove cell debris (0.45 µm filter). Quantitative analysis of 4 MAAs (shinorine, porphyra-334, mycosporine-glycine-alanine, and palythine; see Figure 3) in the clarified extracts was performed using a 1290 series liquid chromatography (LC) system equipped with a 1290 VL pump and a 1260 HiP ALS injector system coupled to a 6460 Triple Quadrupole mass spectrometer (QqQ) (Agilent Technologies), as described by Geraldes et al. [31].

Approximately 5.0 mg lyophilized biomass of each sample, weighed in the precision analytical Ohaus AR3130 balance (Parsippany, NJ, USA), were extracted on ice in 2.0 mL 0.1% (*v*/*v*) formic acid and 0.2 mM ammonium formate (Buffer A), pH ~ 2.55 with disruption by one cycle of sonication (1 min per sample) and vortex mixing (5 s per sample). After 1 h at room temperature (~22 +/− 1 °C), the extracts were centrifuged (10,000 rpm, 10 min, 5 °C) and the supernatant was filtered to remove cell debris (0.45 µm filter). Quantitative analysis of 4 MAAs (shinorine, porphyra-334, mycosporine-glycine-alanine, and palythine; see Figure 3) in the clarified extracts was performed using a 1290 series liquid chromatography (LC) system equipped with a 1290 VL pump and a 1260 HiP ALS injector system coupled to a 6460 Triple Quadrupole mass spectrometer (QqQ) (Agilent Technologies), as described by Geraldes et al. [31].

### 3.5. Phylogenetic Analyses

A maximum likelihood phylogenetic tree was generated using the GTDB-Tk v2.1.0 software [32] and edited via the iTOL phylogenetic trees online tool [33]. Using genome assemblies as input data (either from NCBI or manually added assemblies), this software placed each genome in domain-specific concatenated protein reference trees using 120 bacterial marker genes. Each genome was then allocated to a domain according to the highest ratio of identified marker genes, concatenated in a solo sequence alignment. Taxonomic classification was performed as a combination of GTDB reference tree placements, relative evolutionary divergence (RED), and average nucleotide identity (ANI) compared with reference strains. In addition, specific phylogenies for each gene and one for the complete clustered sequences were constructed using amino acid sequences found in the same strains of cyanobacteria used in the phylogenetic analysis and retrieved from NCBI. Based on the alignment of each amino acid, the phylogenies were assembled in the graphical platform *Phylogeny.fr* [34,35]. Prior to assembly and visualization, an alignment was made using MUSCLE [36] with data curation using Gblocks [37]. Assembly was performed using PhyML [38,39] and tree construction was performed using TreeDyn [40]. Some non-cyanobacterial sequences (*Dactylococcopsis salina* PCC8305, *Porphyra umbilicalis*, and *Chondrus crispus*) known to encode proteins for functional MAA production were included to evaluate the origin and evolutionary history of the cyanobacterial genes [41].

### 3.6. Total Carotenoid Content Determination

A 1.0 mL aliquot was taken from each strain (n = 3) after 45 days of growth and 72 h +/− UVA+UVB light exposure. These aliquots were filtered through a glass microfiber filter (47 mm, GE Life Sciences) under vacuum. Each filter was stored in amber Eppendorf tubes and frozen at −80 °C until extraction, which was performed as described by Strickland and Parson [42]. Total carotenoid content was calculated by the method of Kirk and Allen [43].

## 4. Discussion

The organization of MAA-related genes in the genome of cyanobacteria can occur in diverse configurations, but it seems to be linked to constitutive MAA production yield and gene proximity. The canonical four-gene cluster (*mys*ABCD) is related to the biosynthesis of several MAAs regardless of the biogenic origin of the MAA start unit [44]. The exception was the cluster of *Brasilonema* strains, where production of palythine was found only (*mys*ABCHE, Figure 1). A similar architecture and metabolite production profile was described by Wang et al. [45] in at least seven different organizational groups of bacteriocin gene clusters encoded by 58 cyanobacterial genomes. The gene arrangement correlated to constitutive MAA production in the cyanobacteria strains, which expectedly could be enhanced following UV irradiance or changes to growth media (Table 1 and Table 2). Although the number of strains evaluated was small (n = 10), the annotation highlighted different gene groupings (Figure 1). Only genes arranged in a co-linear cluster resulted in constitutive detection of MAAs, with the fourth interchangeable gene either being a *mys*D, present in *N. spumigena* CENA596 and *S. torques-reginae* ITEP-024, or a combination of *mys*H (dioxygenase) and *mys*E (NRPS-like), found in *B. octagenarum* UFV-OR1 and *B. sennae* CENA114. The presence of the dioxygenase (*mys*H) was recently proposed to be essential for the synthesis of palythine in cyanobacteria, being described in the strain *Nostoc linckia* NIES-25 [46]. The co-linear arrangement of genes organized as a cluster has previously been reported with constitutive MAA production in *Sphaerospermopsis torques-reginae* ITEP-024 [31], *Nostoc punctiforme* ATCC 29133 [47], and *Anabaena variabilis* PCC7937 [7,48]. However, this study is the first report of constitutive MAA biosynthesis in the strain *N. spumigena* CENA596 and induced MAA production in *Brasilonema octagenarum* UFV-OR1 (Table 1 and Table 2).

The enzymes involved in MAA biosynthesis are common and widespread in cyanobacteria, and could be linked to other biosynthetic pathways. Interestingly, the *mys*A+*mys*B and *mys*C+*mys*D fused copies found in *Porphyra umbilicalis* served as an outgroup of the clustered version of the *mys*ABC genes in cyanobacteria, whereas the *mys*D seems to have a divergent origin, being ancestral to the analogous in red algae [49]. The history of this gene cluster is still unclear, with gene transfer between bacteria or dinoflagellate or algae possibly playing a major role in its evolution and occurrence [41]. This could reflect loss of MAA biosynthesis over evolutionary time in these cyanobacteria, with changes in environmental conditions and acclimation. In such a scenario, the genes identified in a non-canonical arrangement and scattered at various loci throughout the genomes most probably encode orthologous proteins.

*Nodularia spumigena* CENA596 and *Sphaerospermopsis torques-reginae* ITEP-024 were the only strains tested that produced detectable concentrations of constitutive MAAs in the same light condition as the others and without additional UV irradiation, described as inductive of MAA production [50,51]. The link between UV irradiance and MAA production has already been vastly reported in the literature [3,9,10,31,50,52,53,54,55,56,57]. This study has provided evidence suggesting the requirement for a co-linearly arranged MAA biosynthetic gene cluster in the production of detectable MAA levels, constitutively and induced by UV light. Conversely, the absence of a co-linear cluster was found to strongly correlate with non-production, even in the presence of known inducers (Table 2). *N. spumigena* CENA596 and *S. torques-reginae* ITEP-024 differ from the remaining strains tested due to their gene cluster arrangement and habitats of origin (Figure 1 and Table 3), isolated from aquatic environments where both can form blooms [17,58]. As filamentous strains, they might also use other UV-avoiding strategies, such as motility and water column migration.

A wide variety of strategies can be employed by cyanobacteria to avoid UV radiation damage apart from photoprotective molecules. Gliding motility and accumulation of carotenoid pigments [59,60], heat dissipation [61], and antioxidant production [62] are some examples. The strains that did not constitutively produce MAAs could be employing one or several other strategies to avoid solar-induced damage. This is particularly relevant for strains isolated in soda lake strains from Pantanal. As discussed by Castenholz and Garcia-Pichel [62], cyanobacteria from hypersaline habitats can cope with the stress of exposure to solar UV radiation by moving upward and downward in the water column during daytime while still capturing optimum light for photosynthesis. This behaviour was described in the motile filamentous *Spirulina labyrinthiformis* [63] in hypersaline mat communities. Apart from *Geminocystis* sp. CENA526, all the other Pantanal strains tested (Table 3) are also filamentous cyanobacteria, which can display this type of motility. It is conceivable that the coccoid *Geminocystis* sp. CENA526 might use carotenoids as alternative photoprotective metabolites since this was the only strain where statistically significant increases in total carotenoid content were detected after UV irradiance (Appendix A).

The presence of presumptive MAA encoding genes in these non-producing strains, that were separated by long intragenic nucleotide sequences (>15.000 bp) [64] might represent relic genomic signatures. This concept has been discussed for genes related to scytonemin biosynthesis [47,65]. In the case of scytonemin genes, these are believed to be non-functional genes as a result of random genetic events such as deletions, insertions or duplications. The genes that displayed a non-canonical arrangement and were scattered throughout the genome in strains isolated from the Pantanal still had motifs consistent with functional *mys* genes (Appendix A). However, they were altered in size or completely absent in some of the strains (Figure 1). If these genes were once organized in a co-linear cluster in ancestral strains, which had subsequently lost gene order conservation, these genes could now possibly be considered relics in the evolution of MAA biosynthesis [62]. Photoprotective mechanisms may now be associated with light-induced behaviour (such as water column migration) and production of other UV-absorbing molecules, such as carotenoids, that are also important in other cellular functions, like light-harvesting for photosynthesis [66].

A combination of factors is relevant when evaluating MAA production in cyanobacteria. As observed in this study, the presence of the cluster and its arrangement seems to matter, as well as physical culturing conditions (photoperiod, pH, temperature) and chemical composition of the media (presence of salt and ammonium), which may cause fluctuations in the detected concentrations of MAAs [48,67]. Hartmann et al. [68] found no detectable MAAs in *Leptolyngbya foveolarum* CCALA081 and *Calothrix* sp. CCALA032, but identified porphyra-334 (0.31 µg mg^−1^ dry weight) in *Nostoc commune* CCALA118. All three strains were grown in BG-11 at 20 °C in a 16:8 h cycle of light/dark. This concentration was greater than what had been described for other strains of the genus *Nostoc*, such as *Nostoc* sp. CCIBt3247 and *Nostoc* sp. CCIBt3292, cultivated in ASM-1 under 24 °C in a 12:12 h light cycle [31] and *Nostoc* sp. R76DM in BG-11 under 25 °C after 72 h of UV irradiation [54]. These differences in MAA levels might be related to culturing variances. The MAA production of these *Nostoc* strains may differ due to evolutionary species-specific variations connected to habitat or evolutionary history (CCALA118 was isolated from soil in the Italian Alps, R76DM from Gujarat, India, and CCIBt3247 and CCIBt3292 from a rainforest in Brazil).

Aigner et al. [69] added interesting data towards correlating MAA production and natural habitat. Analysing three species of *Chamaesiphon,* the authors identified high yields of porphyra-334 in two species isolated from epilithic biofilms from Austrian alpine streams exposed to direct sunlight (*C. starmachii* 1.65 ± 0.68 µg mg^−1^ and *C. geitleri* 0.79 ± 0.15 µg mg^−1^) and a much lower concentration on a species from a shaded stream (*C. polonicus* 0.064 ± 0.013 µg mg^−1^). If habitat is important, then the results obtained in this study for *Brasilonema sennae* CENA114 and *B. octagenarum* UFV-OR1 could also be explained. Both strains were isolated from shaded areas and, despite having the co-linear gene cluster (*mys*ABCHE), did not produce detectable concentrations of MAAs when grown under fluorescent light. This shows that sampling cyanobacteria from diverse and yet underexplored locations is relevant [70]. Searching for strains that are naturally adapted to specific environmental conditions most likely would produce high yields of specialized metabolites associated with the resilience to endure such circumstances [69].

Apart from genetic background, a fully functioning metabolism depends on the nutritional condition that cells are exposed to. Micronutrients, such as trace metals (Cu, Mo, Mn, Zn, and others), are essential for nitrogen fixation, efficient light use, and enzymatic activity [71]. The removal of optimum growth conditions, such as the depletion of phosphorus (P), can alter the metabolic profile of cyanobacteria and cause, for instance, lipid accumulation [72]. Specifically looking at MAAs, researchers have suggested that under low nitrogen (N) there is a reduction in production of these compounds, which are derived from aminocyclohexenone or aminocyclohexenimine scaffolds [73]. However, specific deficiencies, as reported for sulphur, could induce higher yields of some MAAs to counteract stressful conditions [74]. Yet, some cyanobacteria strains can maintain growth and display no significant difference in MAA production under N or P depletion [53].

In this study, MAA biosynthesis was detected when strains were grown in the Z8 medium (Table 1 and Table 2). It has previously been suggested that culture media can influence MAA production, with some media inducing higher MAA yields than others [75]. Medium Z8 is considered a standard medium for growing microorganisms, especially cyanobacteria [76,77,78]. It was of interest to note that of the different culture media tested (ASM-1, BG-11, and Z8), the production of porphyra-334 was outstandingly different particularly in *Nodularia spumigena* CENA596 (considering absolute values). The main finding for the strains tested is that regardless of the nutritional state, without the organized cluster, there is no MAA biosynthesis. Therefore, genetic background, nutritional environment, and physical inducing agents all act in concert to explain MAA biosynthesis in cyanobacteria.

Medium ASM-1 is considered nutritionally poor, with lower nitrogen and phosphorus concentrations when compared with others (Appendix A). Geraldes et al. [31] did not detect either shinorine or porphyra-334 in 21 strains of cyanobacteria cultivated in ASM-1, even after 72 h of UVA and UVB light exposure. Either the nutritional status provided by this medium was not sufficient to support MAA biosynthesis, or the lack of production reflected the genetic organization of the MAA biosynthetic genes which was not determined in this study. The authors estimated MAA production in *N. spumigena* CENA596 and *S. torques-reginae* ITEP-024 growing in ASM-1, which were two of the strains used in the study reported herein. Differing levels of shinorine and porphyra-334 were reported in the two studies. This variation illustrated the importance of physical cultivation parameters (temperature, light cycle, and inoculum) that can influence the final yield of metabolites. For strains such as *Geminocystis* sp. CENA526, with no co-linear cluster arrangement (Figure 1), growth in different media (ASM-1, Z8, and BG-11) was not sufficient to complement the non-canonical structure of the MAA genes and hence failure to detect constitutive shinorine or porphyra-334 (Table 1). A more nutritionally rich cultivation medium for cyanobacteria is BG-11, being particularly high in concentrations of N derived from nitrates and ammonium [23], which are a supplement to the medium (Appendix A). Porphyra-334 is biosynthesized from the condensation of threonine to mycosporine-glycine. Shiio and Nakamori [79] studied the production of threonine in bacteria, describing some nutritional aspects that could affect the amount of threonine synthesized. Significantly higher threonine yields were achieved in the presence of ammonium, which directly feeds into the aspartic acid pathway. The differences measured in the cyanobacteria grown in this study may be linked to this enriched environment provided by BG-11 (Table 1). As for shinorine, no significant difference in production titres was observed when the strains were grown in different media, with absolute values varying less than 1.4 times between the highest and lowest values measured in the same strain (Table 1). This suggests additional studies are required to fully elucidate the biosynthetic pathway leading to shinorine production.

Stress factors may induce the synthesis of MAAs even in strains without an evident constitutive production. The Pantanal strain *Anabaenopsis elenkinii* CCIBt3563 tested herein had no detectable levels of shinorine or porphyra-334 when grown in medium Z8(mod). However, Bairwa et al. [56] reported 0.1 µg mg^−1^ dry weight production of shinorine in a UVB-light-induced BG-11 culture of *Anabaenopsis* sp. SLCyA isolated from a hypersaline lake in Sambar (India). Although the two strains belong to the same genera and were isolated from similar habitats (soda lakes), other factors such as the genetic arrangement of the biosynthetic genes might be important to consider when comparing levels of MAA production. Genome sequences for *Anabaenopsis* strains are underrepresented in databases and further taxon sampling is warranted to fully appreciate the genetic basis of MAA biosynthesis in these habitat-specialized cyanobacteria.

The associations between an organized co-linear gene cluster arrangement and constitutive MAA biosynthesis indeed makes sense, especially when considering the evolutionary relationships established in both the phylogenetic tree (Figure 2) and phylogeny of each gene (Appendix A). Most MAA-producing cyanobacteria reported in the literature are Nostocales strains [7,14,31,47,48,53,54,55,56,67,68,74,80]. Nevertheless, some genera from Chroococcales, Oscillatoriales, and Synechococcales, such as *Microcystis* [12,81], *Oscillatoria* [44], and *Chamaesiphon* [69], have also been described as MAA producers. In several strains taxonomically distinct from Nostocales (apart for *Microcystis*), the presence of the co-linear gene cluster was not reported, probably due to a paucity of genomic data. The widespread presence of the co-linear cluster in Nostocales could indicate that the common ancestor of this taxonomic order demonstrated the same gene order conservation (Figure 2). However, due to the rarity and considerable diversification of the cluster and the identification of MAA production in strains from other orders, it seems more parsimonious to assume that the organized co-linear arrangement occurred in specific strains or in branches within the phylum (via conjugational transference, genomic rearrangements, or other genetic mechanisms) rather than being considered a common feature for all cyanobacteria that was later lost in most of the branches. The continuous isolation and sequencing of cyanobacterial strains from different orders and families will produce data that could further clarify this topic.

When analysing the results of this study, it was important to consider that the strains were long-term monocultures grown in laboratory settings, in media providing high nutrient availability and controlled light exposure and temperature. This greatly contrasts with environmental conditions that impose biotic and abiotic stresses, changing gene expression, and metabolomics [82]. Lakeman et al. [83] described that fixed laboratory settings gradually alter natural traits of the phytoplankton observed in the field, such as filament morphology and toxin-producing capacity. To some extent, this effect could also be occurring in MAA production. To test this hypothesis, studies focusing on transcriptomes and metabolomes of freshly isolated cultures could generate important data.

## 5. Conclusions

MAA production was quantified in extracts of 10 cyanobacterial strains using LC/MS-MS against 4 known MAA standard compounds, and qualitatively by UV-visible absorption spectroscopy. MAAs were only produced in strains where the genes necessary for biosynthesis were arranged in co-linear architectures. MAA production was constitutive in these strains and the concentration of MAAs unsurprisingly increased significantly upon UV light exposure. Constitutive MAA production was not detected in cyanobacteria where MAA biosynthetic genes displayed non-canonical arrangements, often separated by lengthy intragenic regions. Surprisingly, production of MAAs could not be induced in these strains either by UV irradiation or by changing the growth media. However, some of these non-MAA-producing strains contained larger quantities of carotenoid pigments than the MAA producers, suggestive of alternative photoprotection strategies in these cyanobacteria. A wider appreciation of how gene order conservation is related to the biosynthesis of photoprotective compounds is necessary to understand the ecology of cyanobacteria.

## Figures and Tables

**Figure 1 molecules-28-01420-f001:**
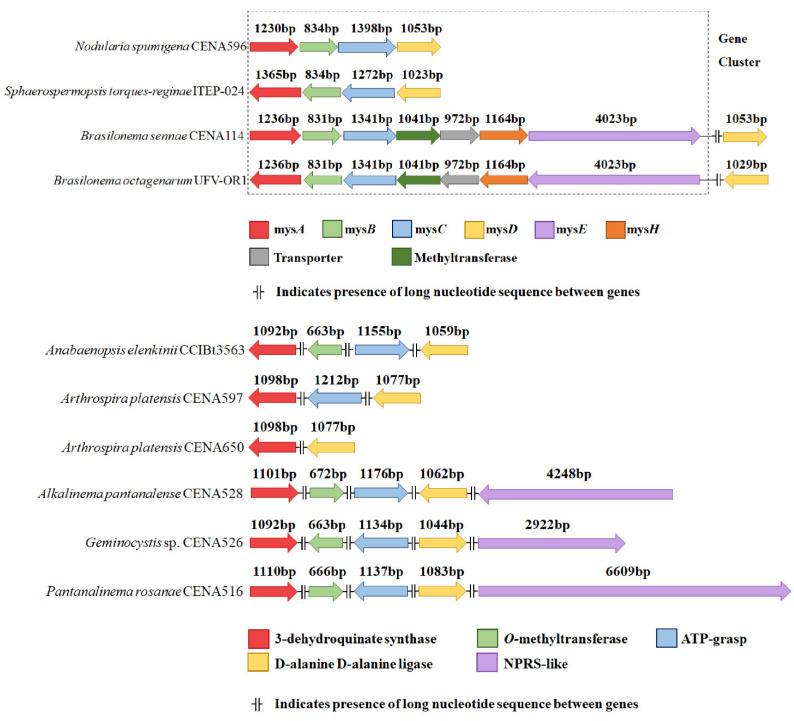
Genes identified at the draft genomes of each strain studied, indicating the size of each fragment (in base pairs) and the 5′→3′ direction according to each gene orientation. The genes identified outside a co-linear cluster arrangement are identified by the expected encoded protein. More detailed information describing the homology between genes is presented in the Appendix A.

**Figure 2 molecules-28-01420-f002:**
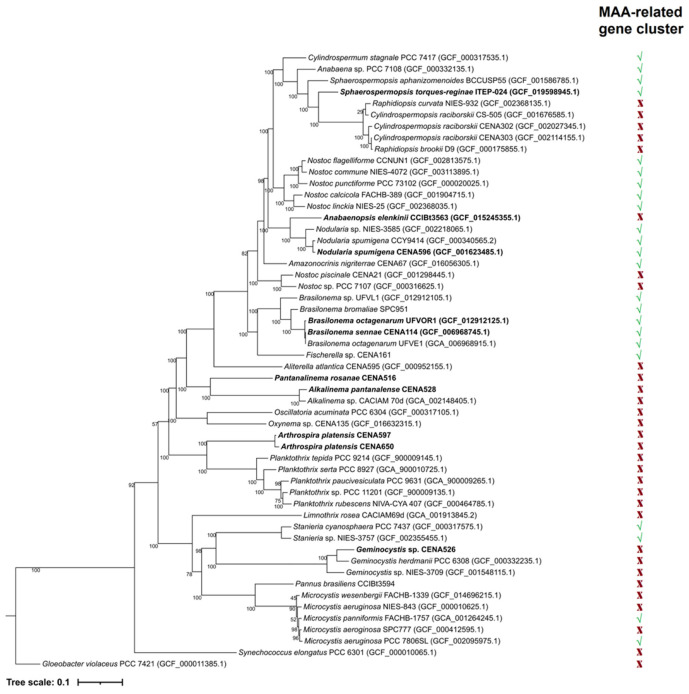
Maximum likelihood phylogenetic tree based on 120 cyanobacterial single-copy conserved markers displaying the relationship of the studied strains (in bold) with other cyanobacterial genomes. In the right parallel column, (√) indicates the presence and (X) indicates the absence of the MAA-related gene cluster for each strain.

**Figure 3 molecules-28-01420-f003:**
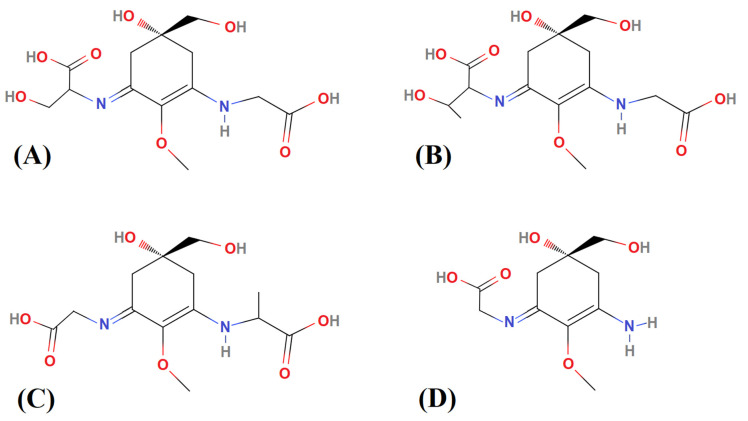
Chemical structure of each MAA standard used for quantitative analyses. (**A**) Shinorine, (**B**) porphyra-334, (**C**) mycosporine-glycine-alanine, and (**D**) palythine.

**Table 1 molecules-28-01420-t001:** Concentration of shinorine (SH) and porphyra-334 (P-334) detected in cyanobacteria strains according to the culture medium.

Strain	Culture Medium	SH (µg mg^−1^)	P-334 (µg mg^−1^)	Palythine	Myc-Gly-Ala
*Nodularia spumigena* CENA 596	Z8 ASM-1 BG-11	0.062 ± 0.036 0.046 ± 0.018 0.066 ± 0.037	4.791 ± 3.639 2.586 ± 1.579 6.169 ± 4.559	- - -	- - -
*Sphaerospermopsis torques-reginae* ITEP-024	Z8 ASM-1 BG-11	3.143 ± 0.983 2.764 ± 1.322 4.068 ± 1.467	0.319 ± 0.179 0.256 ± 0.095 0.356 ± 0.188	X - X	- - X
*Geminocystis* sp. CENA 526	Z8 ASM-1 BG-11	- - -	- - -	- - -	- - -
*Arthrospira platensis* CENA 597	Z8	-	-	-	-
*Arthrospira platensis* CENA 650	Z8	-	-	-	-
*Pantanalinema rosanae* CENA 516	Z8	-	-	*-*	-
*Alkalinema pantanalense* CENA 528	Z8	-	-	*-*	-
*Anabaenopsis elenkinii* CCIBt 3563	Z8 modified	-	-	-	-
*Brasilonema sennae* CENA 114	Z8 (0)	-	-	X	-
*Brasilonema octagenarum* UFV-OR1	Z8 (0) BG-11	- -	- -	X X	- -

Results presented as µg _MAAs_/mg _biomass_ with standard deviation_._ (-) indicates no MAAs detection. (X) indicates the presence of MAAs below the detection limit (<0.01). Results are based on duplication of biological triplicates (i.e., n = 6) with raw data provided in the Appendix A.

**Table 2 molecules-28-01420-t002:** Concentrations of shinorine (SHI) and porphyra-334 (P-334) detected in the cyanobacteria studied according to the UV irradiance setting.

Strain	Setting	SH (µg mg^−1^)	P-334 (µg mg^−1^)	Palythine	Myc-Gly-Ala
*Nodularia spumigena* CENA596	Control 72 h after UV	0.066 ± 0.003 0.066 ± 0.002	2.353 ± 0.043 * 7.658 ± 0.451 *	- X	- -
*Sphaerospermopsis torques-reginae* ITEP-024	Control 72 h after UV	3.849 ± 0.132 * 9.032 ± 1.121 *	0.227 ± 0.059 * 1.699 ± 0.127 *	X 0.034 ± 0.005	- X
*Brasilonema octagenarum* UFV-OR1	Control 72 h after UV	- -	- -	X 0.278 ± 0.039	- -
*Brasilonema sennae* CENA114	Control 72 h after UV	- -	- -	X X	- -
*Geminocystis* sp. CENA526	Control 72 h after UV	- -	- -	- -	- -
*Alkalinema pantanalense* CENA528	Control 72 h after UV	- -	- -	- -	- -
*Anabaenopsis elenkinii* CCIBt3563	Control 72 h after UV	- -	- -	- -	- -
*Arthrospira platensis* CENA597	Control 72 h after UV	- -	- -	- -	- -
*Arthrospira platensis* CENA650	Control 72 h after UV	- -	- -	- -	- -
*Pantanalinema rosanae* CENA516	Control72 h after UV	- -	- -	- -	- -

Results are presented as μg_MAAs_/mg_biomass_ with standard deviation. (-) indicates no MAAs detection. (X) indicates the presence of MAAs below the detection limit (<0.01). (*) indicates a significant statistical difference between control and 72 h after UV irradiance based on the ANOVA test (*p* < 0.05). Results are based on biological triplicates (i.e., n = 3) with raw data provided in the Appendix A.

**Table 3 molecules-28-01420-t003:** Description and isolation details of each strain used in this study.

Strain	Culture Medium	Habitat of Origin	Genome Assembly Accession Number	Description and Isolation Reference
*Nodularia spumigena* CENA596	Z8	Shrimp production pond (32°12′19″ S, 52°10′42″ W, Rio Grande, Rio Grande do Sul/Brazil)	GCA_001623485.1	[19]
*Sphaerospermopsis torques-reginae* ITEP-024	Z8	Freshwater Tapacurá reservoir (8°02′14″ S 35°09′46″ W, Recife, Pernambuco/Brazil)	GCA_019598945.1	[20]
*Geminocystis* sp. CENA526	Z8	Saline-alkaline lake (19°26′24″ S, 56°05′58″ W, Aquidauana, Mato Grosso do Sul/Brazil)	KF246492 (only 16S RNA available)	[21]
*Arthrospira platensis* CENA597	Z8	Saline-alkaline lake (19°27′3.13″ S, 56°7′42.19″ W, Aquidauana, Mato Grosso do Sul/Brazil)	Unpublished	Unpublished
*Arthrospira platensis* CENA650	Z8	Saline-alkaline lake (19°22′47.2″ S, 56°18′51.6″ W, Aquidauana, Mato Grosso do Sul/Brazil)	Unpublished	Unpublished
*Pantanalinema rosanae* CENA516	Z8	Saline-alkaline lake (19°28′13″ S, 56°03′22″ W, Aquidauana, Mato Grosso do Sul/Brazil)	KF246483 (only 16S RNA available)	[21]
*Alkalinema pantanalense* CENA528	Z8	Saline-alkaline lake (19°26′56.0″ S, 56°07′54.8″ W, Aquidauana, Mato Grosso do Sul/Brazil)	KF246494 (only 16S RNA available)	[21]
*Anabaenopsis elenkinii* CCIBt3563	Z8_mod._ *	Saline-alkaline lake (18°57′35″ S, 56°37′18″ W, Corumbá, Mato Grosso do Sul/Brazil)	GCA_015245355.1	[22]
*Brasilonema sennae* CENA114	Z8_0_ **	Iron water pipe from spring water (23°46′ S, 46°18′ W, Paranapiacaba, São Paulo/Brazil)	GCA_006968745.1	[23]
*Brasilonema octagenarum* UFV-OR1	Z8_0_ **	Orchid leaf axil (20°20′ S, 41°08′ W, Venda Nova do Imigrante, Espírito Santo/Brazil)	GCA_012912125.1	[24]

* With added NaCl. ** Without nitrogen sources.

## Data Availability

The MAA data presented in this study are available in this Google Drive link.

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
