# Peer review of "Exploring the Relationship between Biosynthetic Gene Clusters and Constitutive Production of Mycosporine-like Amino Acids in Brazilian Cyanobacteria"

_molecules, 2023, doi:10.3390/molecules28031420_

Round 1

Reviewer 1 Report

The authors presented MAAs production in 10 cyanobacterial strains and discussed ecological relationship between gene order and biosynthesis of MAAs. This reviewer appreciate the importance of the study and largely agree with the their discussion. I only have a few comments that may help to improve the quality of this manuscript.

- Line 106: "e-value >= 1.10-5" does not seem to be right. Please correct this.

- Figure 2: Information for "genomic fragments" and homology among the same genes are missing. It would be great if the authors could present a new figure showing them.

- Figure S7-S10: The authors presented phylogenetic trees for each of genes related to MAA biosynthesis. It would be great if the authors could present a tree of concatenated sequences of the key genes (just like the multi-locus sequence typing method) and compare it with the genomic relatedness. The authors may discuss more about evolutionary relationships by comparing a concatenated tree and a species tree (Figure 3).

Reviewer 2 Report

The MS is generally well-organized. I only suggest two minor issues need to be modified .

(1) Please consider to modify the titles of tables to make more concise. Maybe some footnotes can be added. 

(2) References: please correct the format of some references.

Author Response

Reviewer #2
Comment 1: Please consider to modify the titles of tables to make more concise. Maybe some footnotes can be added.
Response: We agree with the reviewer and have altered the titles and legends for tables 2 and 3 in order to make it more concise (page 7 line 202 and page 8 line 222).
Comment 2: References: please correct the format of some references.
Response: The formats of all references now follow the correct journal format. Please see references 1, 6, 8, 15, 31, 44, 73 and 74 (see pages 14 to 16 from line 472 to 626).

Reviewer 3 Report

Comments to the Authors

Authors have isolated ten UV-exposed cyanobacterial strains from various Brazilian habitats and tested them in respect of their MAA content under normal illumination and also after UV exposure. Two of these ten strains showed remarkable shinorine and porphyra-334 content; the authors explained this phenomenon by the genome structure of these strains. Although the topic is of general interest, the quality of the manuscript suffers from many major flaws.

Specific comments:

1) First of all, the major conclusion of the manuscript is not well supported by data: i) although the structure of the two Brasilionoma strains shows similar gene structure to that of Nodularia and Sphaerospermopsis, they contains only traces of MAAs, ii) the conclusion is based on results only of two strains which number is very low to get any solid conclusions.

2) Besides fresh isolates, the work seems to be completed with some commercially available strains (e.g. Nostoc punctiforme) with reportedly high MAA levels.

3) The structuring of the manuscript is entirely wrong. A short Introduction and, more importantly, a very short Results (less than one page) is followed by a very long discussion (about 5 pages) which is nonsense.

4) Very critically, the MAA contents in Table 1 and Table 2 (control lines) are entirely different which make all results questionable.

Minor things, typos

L14 The abb EVS is wrongly solved in the Abstract

L15 encoding

L20 and elsewhere UV light

L33 cyanobacterial

L37 UVB and UVA region rather than UVB and UVA instead of UVB and UVA spectrum. Give wavelengths for general readers.

L47/48 improve grammar

L56 the word “ideal” does not seem appropriate

L57/58 four genes are mentioned but only three are listed. The word “thus” is inappropriate.

L82 “recovered” apparently wrong wording

L88/89 Philips

L100/101. The reported Anabaena is not shown in Fig. 3 and the Nostoc strain is also different at these two locations.

L105 resolve the abb NCBI

L120 “5 mg +- 0.01. Correct

L124 “5˘C” (no space)

Fig. 1. Use larger fonts for the symbols of elements.

L154. A filter with a diameter of 47 mm is two large to filter 1 mL of cell suspension. 25 mm filters seem more appropriate.

L180 “although” inappropriate word

L196 “As expected”

Author Response

Reviewer #3
Specific comments:
Comment 1: First of all, the major conclusion of the manuscript is not well supported by data:
i) although the structure of the two Brasilionoma strains shows similar gene structure to that of Nodularia and Sphaerospermopsis, they contain only traces of MAAs, ii) the conclusion is based on results only of two strains which number is very low to get any solid conclusions.
Response: The manuscript has been revised to refrain from using active tense and to concentrate more on discussing how the data provided by only two strains can convincingly support our observations and conclusions - that canonical gene order conservation is simingly essential for constitutive MAA biosynthesis; although we fully acknowdle that detectable levels of MAAs do require, in some circustances (e.g. Brasilonema cotagenarum), induction, either by exposure to UV irradiance or growth in differente media. We thank the referee for making this observation and we have modified the text accordingly (page 10 lines 271-273 and 289-292).
Comment 2: Besides fresh isolates, the work seems to be completed with some commercially available strains (e.g. Nostoc punctiforme) with reportedly high MAA levels.
Response: The referre is incorrect. All strains used in this study were isolated and maintained at CENA. No strains were obtained commercially hence detection of high levels of MAA production reflects a natural phenomenon from strains isolated from Tropical environments with yearlong high solar UV exposure. On the contrary, if commercial strains were used, that might have been in culture for many years or decades, one might expect a natural response (production of MAA natural photo protective compounds) to UV radiation to be diminuished or lost completely. We have actually considered loss of adaptive traits following prolonged growth under laboratory conditions in the discussion (page 13 line 441), which we support with reference 83.
Comment 3: The structuring of the manuscript is entirely wrong. A short Introduction and, more importantly, a very short Results (less than one page) is followed by a very long discussion (about 5 pages) which is nonsense.
Response: My co-authors and I have many years of research experience and writing scientific manuscripts. Our combined styles of writing are to present the readers with concise Introduction and Results sections. This is in agreement with Guidelines To Authors given by Molecules. We do agree with the referre that our discussion is lengthy and considering referee’s comment 1 and 3, we have edited the discussion so that we concentrate solely on describing results obtained in the experiments and have refrained from making assumptions based beyond our data. Large sections of the Discussion text have therefore been removed (examples: lines 273 to 281 from page 10 and lines 339 to 350 in page 15). Substantial proportions of the Discussion text have been removed (around 300 words) and we now fell that the text is more concise and reads better. We thank the reviewer for suggesting a structural rearrangement of the manuscript.
Comment 4: Very critically, the MAA contents in Table 1 and Table 2 (control lines) are entirely different which make all results questionable.
Response: We acknowledge the concerns of the reviewer, but in our experience, these quantitative values obtained by very sensitive mass spectrometry measurements fall in expected variations. In our hands, the largest variation in quantifying MAA biosynthesis isn’t as one would expect from growing cyanobacteria, but actualy comes from between batch variation from the defined growth media and culture conditions (for example fluctuations in light intensity). This variation has been reported previously by Geraldes et al. in their groundbreaking analytical studies on MAA biosynthesis using mass spectrometry (please see references 31 and 44). We believe that it is important to present such variation and focus on the idea that despite these fluctuations, both strains produced detectable concentrations of MAAs every time they were evaluated.

General remarks from reviewer 3
Comment 5: L14 The abb EVS is wrongly solved in the Abstract
Response: The correction has been made on page 1 line 14.
Comment 6: L15 encoding
Response: The correction has been made on page 1 line 15.
Comment 7: L20 and elsewhere UV light
Response: The text has been ammended throughtout to explain UV more correctly as either exposure, light, irradiance or irradiation (please see page 1 lines 20, 22 and 34; page 2 lines 52 and 71; page 3 line 92; page 7 lines 212 and 218; page 10 lines 287 and 299; page 11 lines 313 and 339).
Comment 8: L33 cyanobacterial
Response: The correction has been made on page 1 line 34.
Comment 9: L37 UVB and UVA region rather than UVB and UVA instead of UVB and UVA spectrum. Give wavelengths for general readers.
Response: The correction has been made on page 1 lines 38/39.
Comment 10: L47/48 improve grammar
Response: The sentence has been rewritten to improve the grammar (page 2 line 47-49).
Comment 11: L56 the word “ideal” does not seem appropriate
Response: The word was substituted using “expected”, which we fell is a justified adjective because we also provide references supporting this gene arrangement (page 2 line 57).
Comment 12: L57/58 four genes are mentioned but only three are listed. The word “thus” is inappropriate.
Response: The sentence has been rewritten (page 2 lines 60/61).
Comment 13: L82 “recovered” apparently wrong wording
Response: The word has been substituted with “concentrated” (page 2 line 84).
Comment 14: L88/89 Philips
Response: The correction has been made (page 3 lines 90/91).
Comment 15: L100/101. The reported Anabaena is not shown in Fig. 3 and the Nostoc strain is also different at these two locations.
Response: The strain of Anabaena used to search for the genes was not included in the phylogenomic tree of figure 3 due to its recent modified taxonomic designation (changing to another genus). We felt that the taxonomic placement would be inappropriate to include in the phylogeny because of a posertive of genetic information from species in the new genus. However, the loss of this Anabaena strain was compensated by the inclusion of other Anabaena strain in the phylogeny as shown in figure 3 (page 9). The Nostoc punctiforme used for gene search is also included in figure 3, however it has a new strain code (PCC 73102). Both strain codes are now written in the main text (page 3, line 104/105). The newer code (PCC 73102) is provided in figure 3 (page 9).
Comment 16: L105 resolve the abb NCBI
Response: The abbreviation was justified when first used, on page 3 Line 103.
Comment 17: L120 “5 mg +- 0.01. Correct
Response: The symbol has been substituted throughtout the manuscript (please see page 2 line 76; page 4 lines 123 and 126).
Comment 18: L124 “5˘C” (no space)
Response: The correct format has been applied throughtout the text (please see page 2 lines 76 and 84; page 4 lines 126 and 127; page 5 line 161; page 11 lines 336, 338 and 339).
Comment 19: Fig. 1. Use larger fonts for the symbols of elements.
Response: The suggested format change to the font in figure 1 have been made (page 4).
Comment 20: L154. A filter with a diameter of 47 mm is two large to filter 1 mL of cell suspension. 25 mm filters seem more appropriate.
Response: In our laboratory we handle various culture volumes and for logistical reasons (e.g. cost) all members of the laboratory use the same diameter filter and filtration apparatus. Additionally, larger filter ensures no loss of biomass, particularly when handling smaller volume.
Comment 21: L180 “although” inappropriate word
Response: The word has been substituted with “while” (page 6 line 191).
Comment 22: L196 “As expected”
Response: The substitution has been made (page 7 line 207).

Round 2

Reviewer 3 Report

Comments to the Authors

Dextro et al. significantly improved their manuscripts and sufficiently responded some Comments to the Authors. However, the manuscript still suffers from several major flaws.

Specific comments:

1) Most critically, there is a major contradiction in the Authors’ response to comment 4 (“Very critically, the MAA contents in Table 1 and Table 2 (control lines) are entirely different which make all results questionable.”)

claiming that

“the largest variation in quantifying MAA biosynthesis … comes from between batch variation from the defined growth media and culture conditions (for example fluctuations in light intensity). This variation has been reported previously by Geraldes et al. in their groundbreaking analytical studies on MAA biosynthesis using mass spectrometry (please see references 31 and 44). We believe that it is important to present such variation and focus on the idea that despite these fluctuations, both strains produced detectable concentrations of MAAs every time they were evaluated.”

and the content of the manuscript: Tables 2 and 3 show about 3-fold difference in SH and about 9-fold difference in P-334 content (under the same conditions) with remarkably low experimental errors which do not match these very high variations at all! One possible explanation for this that rather than from biological replicates, Author calculated experimental error only from technical replicates. However, especially in cases with such high variations, only experimental errors calculated from (sufficient number of) biological replicates make any sense. In other words, more repetitions is needed to get reliable results and conclusions. (Verification and reproducibility is a very important issue in science – one experiment is not enough!)

2) At several(!) locations, the experimental error is provided in a way which is completely against scientific convention. Normally, the accuracy of the given value and the experimental error should have the same(!) accuracy. E.g. 0.17 ± 4.25 × 10-5 is entirely wrong! 0.17 ± 0.0001 also, etc. Nevertheless, the kind of experimental error (standard error, standard deviation) should also be given.

3) L126. It is unbelievable that biomass with such a high accuracy (5 ± 0.01 mg) was obtained with the extraction process given. (By the way this number is also wrongly given! Correctly, it should be expressed as 5.00 ± 0.01.)

4) After some shortening, the Discussion is still disproportionally long.

Author Response

Reviewer #3

Specific comments:

Comment 1) Most critically, there is a major contradiction in the Authors’ response to comment 4 (“Very critically, the MAA contents in Table 1 and Table 2 (control lines) are entirely different which make all results questionable.”)

claiming that

“the largest variation in quantifying MAA biosynthesis … comes from between batch variation from the defined growth media and culture conditions (for example fluctuations in light intensity). This variation has been reported previously by Geraldes et al. in their groundbreaking analytical studies on MAA biosynthesis using mass spectrometry (please see references 31 and 44). We believe that it is important to present such variation and focus on the idea that despite these fluctuations, both strains produced detectable concentrations of MAAs every time they were evaluated.”

and the content of the manuscript: Tables 2 and 3 show about 3-fold difference in SH and about 9-fold difference in P-334 content (under the same conditions) with remarkably low experimental errors which do not match these very high variations at all! One possible explanation for this that rather than from biological replicates, Author calculated experimental error only from technical replicates. However, especially in cases with such high variations, only experimental errors calculated from (sufficient number of) biological replicates make any sense. In other words, more repetitions is needed to get reliable results and conclusions. (Verification and reproducibility is a very important issue in science – one experiment is not enough!)

Response: We acknowledge the concerns of the referee about additional replicates. However, we feel that the referee has misunderstood our methodology. Our reading of the referee’s comments suggests to us that we had taken technical replicates from a single sample. We would like to emphazise to the referee that this was not the case. We actually performed triplicate experiments (i.e. provided biological replication for each experimental variable). We added this information in the text (PAGE 7 LINE 208; PAGE 8 LINE 230). We maintain cultures growing constantly in the laboratory, which has allowed us to rapidly obtain new data and recalculate our values over the last few weeks, which increased the number of biological replicates to n = 6. This new data is given in the Supplementary file (PAGES 21-25) and alterations to the text were made in the manuscript (PAGE 6 LINES 190/191; 193-195; PAGE 7 LINES196-199; PAGE 12 LINES 378-380 and LINES 407-410). This new data reaffirms our previous findings and conclusions whilst addressing the concerns of the referee in terms of standard deviation and titres of the control measurement.

Comment 2) At several(!) locations, the experimental error is provided in a way which is completely against scientific convention. Normally, the accuracy of the given value and the experimental error should have the same(!) accuracy. E.g. 0.17 ± 4.25 × 10-5 is entirely wrong! 0.17 ± 0.0001 also, etc. Nevertheless, the kind of experimental error (standard error, standard deviation) should also be given.

Response: My co-authors and I would like to thank the referee for this comment, improving in the correct display of our data according to scientific convention. Data values plus each corresponding standard deviation have been rewritten in the values displayed on Table 2 (PAGE 7) and Table 3 (PAGE 8).

Comment 3) L126. It is unbelievable that biomass with such a high accuracy (5 ± 0.01 mg) was obtained with the extraction process given. (By the way this number is also wrongly given! Correctly, it should be expressed as 5.00 ± 0.01.)

Response: We acknowledge the concerns of the reviewer and understand that Materials and Methods need to be written in a way that are reproducible for others to follow. We have therefore amended the text to “approximately 5.0 mg” rather that the high accuracy previously written and we also added the model of precision balance used (PAGE 4 LINE 124/125).

Comment 4) After some shortening, the Discussion is still disproportionally long.

Response: We acknowledge the concerns of the reviewer, but without indication of specific parts regarded as “unnecessary” by the referee, we feel that further shortening the Discussion only serves word reduction for the sack of a shorter text. Referees #1 and #2 had no comments regarding the length or content of the discussion. We therefore feel strongly that any attempt to shorten the discussion will severely detract from our interpretation of the scientific data. We ask the Academic Editor to adjudicate on this matter.